# Process Development for Batch Production of Micro-Milling Tools Made of Silicon Carbide by Means of the Dry Etching Process

**DOI:** 10.3390/mi14030580

**Published:** 2023-02-28

**Authors:** Christian-G. R. Wittek, Lukas Steinhoff, Selina Raumel, Michael Reißfelder, Folke Dencker, Marc C. Wurz

**Affiliations:** 1Institute of Micro Production Technology (IMPT), Leibniz University Hanover, 30823 Garbsen, Germany; 2Reißfelder Profilschleifen GmbH, 75031 Eppingen, Germany

**Keywords:** silicon carbide, reactive ion etching, photolithography, batch production, micro-milling

## Abstract

Downsized and complex micro-machining structures have to meet quality requirements concerning geometry and convince through increasing functionality. The development and use of cutting tools in the sub-millimeter range can meet these demands and contribute to the production of intelligent components in biomedical technology, optics or electronics. This article addresses the development of double-edged micro-cutters, which consist of a two-part system of cutter head and shaft. The cutting diameters are between 50 and 200 μm. The silicon carbide cutting heads are manufactured from the solid material using microsystem technology. The substrate used can be structured uniformly via photolithography, which means that 5200 homogeneous micro-milling heads can be produced simultaneously. This novel batch approach represents a contrast to conventionally manufactured micro-milling cutters. The imprint is taken by means of reactive ion etching using a mask made of electroplated nickel. Within this dry etching process, characteristic values such as the etch rate and flank angle of the structures are critical and will be compared in a parameter analysis. At optimal parameters, an anisotropy factor of 0.8 and an etching rate of 0.34 µm/min of the silicon carbide are generated. Finally, the milling heads are diced and joined. In the final machining tests, the functionality is investigated and any signs of wear are evaluated. A tool life of 1500 mm in various materials could be achieved. This and the milling quality achieved are in the range of conventional micro-milling cutters, which gives a positive outlook for further development.

## 1. Introduction

The miniaturization of conventional micro-machining components goes hand in hand with the analogous miniaturization of the tools, which have to meet even higher demands. This requires innovations on the way to downsized and complex structures of high quality and comprehensive functionality. Microsystem-technology is steadily gaining ground. Efficient processes are the basis for the production and further development of micro-tools. Possible application scenarios include the production of microfluidic or biological systems [1,2,3,4]. The demand for a functional surface to reduce friction and wear or to modify special material properties is becoming increasingly greater and is reflected in the increase in the efficiency of the component [5]. The production of MEMS by means of micro-milling in the fields of communication and optics is growing [6,7,8]. Small tolerances and surface roughness need to be achieved [9].

In microsystems technology, these tools enable the production of structures in the sub-millimeter range down to a few microns. The reduction of the tool diameter is accompanied by cutting edge rounding and a possible reduction in stiffness, which must be taken into account when designing such tools [9]. Conventionally available micro-milling cutters are manufactured individually, which is reflected in the high material and time-contingent and resulting costs [10].

The hardness of silicon carbide (SiC) means it is very well suited as a material for milling in order to keep material wear low [11]. In contrast to conventionally manufactured milling cutters, this approach with SiC offers two significant advantages. With the new concept, the milling cutter consists of the solid ceramic material silicon carbide. The second advantage is batch production, i.e., simultaneous production of a large number of components instead of milling each cutter from the carbide [12].

The aim of this work is the process development of silicon carbide micro-milling heads via batch production. The milling heads are manufactured by means of the dry etching process. In previous work, silicon carbide milling cutters could already be produced through a deep reactive ion etching process with diameters of 400 µm and a structural height of 66 µm [13,14,15]. The process was performed on a 2 inch silicon carbide substrate to manufacture up to 1200 milling heads. This established process involving reactive deep ion etching is to be adapted to a process involving reactive ion etching.

Moderate etch rates and further miniaturization to structure diameters down to 50 µm are the targets to be achieved. The number of simultaneously manufactured milling heads is also to be increased fourfold to reduce the cost per unit. The focus of this work is on investigating a suitable process to realize these requirements. This process covers the initial state of a 4” substrate up to the functional milling tool. The described goals require a new mask design, which is being developed by means of photolithography. This lithographic process and the followed steps of electroplating, ion beam etching and reactive ion etching are developed. Back-end processes such as dicing and joining complete the process. In the final milling tests, the functionality will be evaluated and the wear cases will be recognized and classified.

The novel approach to the production of micro-millers contributes to the more economical production of micro-millers. In addition, the developed process is a pioneer of the three-dimensional structuring of silicon carbide, which goes beyond the structural dimensions of semiconductor manufacturing.

## 2. Materials and Methods

The milling tool is realized as a two-part system consisting of a milling cutter shaft and a milling cutter head, which are joined in the final step.

The milling cutter shank is made of the aluminum alloy AlMg1. This has a cylindric tip with a milled pocket, which serves as a centering aid for the milling heads. The shank has a diameter of 3 mm, and the tip of the shank has a diameter of 1.4 mm. The pocket has an internal dimension of 1.1 mm to accommodate milling heads with an edge length of 1 millimeter. There is a cross-shaped groove at the level of the pocket, which enables the even distribution of the joining medium. The use of aluminum is possible due to the low forces during micro-milling. It also offers comparatively easier machinability than steel. A scheme for this system is shown in Figure 1.

The milling head has a plateau with the structured milling tip, and it has two cutting edges. The head is produced in a batch using the dry etching process. The material used is single-crystal silicon carbide. By reducing the dimensions of the cutter geometry, different material properties in the grain structure have a greater influence. For this reason, a carbide substrate with a fine to ultra-fine grain structure is often selected when miniaturizing conventional milling cutters. Alternatively, this effect is omitted, since there is only one crystal orientation in the single crystal and no grain boundaries exist, but there is increasing anisotropy of the material properties. The second reason to use silicon carbide is due to its mechanical properties and the available processing technologies.

### 2.1. Metallization and Photolithography

In this work, a silicon carbide single-crystal structure 4H with a diameter of 4 inches and a thickness of 500 µm is used as a substrate.

The metallization for full-surface electrical contact for later electroplating is applied via sputter deposition. At first, sputter etching is performed to remove organic pollutions and native oxide layers. A 50 nm thick layer of adhesion-promoting chromium layer followed by a 500 nm thick permalloy seed layer for electroplating (Figure 1i) are applied. The permalloy has a similar crystal grating to nickel and is therefore suitable for subsequent electroplating.

The metallization is followed by coating with a positive photoresist and its patterning via photolithography (Figure 1ii). For performing the photolithography, an exposure mask is produced. Many individual tests are planned for the investigation of the optimum etching parameters and various influencing variables. These are carried out at the chip level. The test structures and different cutter sizes on each of these chips enable the easier validation of the etch depth and anisotropy and appraisal of the optimal parameters for each diameter. The chips have a dimension of 10 mm edge width. This means that 52 chips can be accommodated on 1 wafer. On these chips, an array of ten-by-ten cutters is planned, each with an area of one square millimeter. This means that 100 milling heads can be produced on 1 chip, which corresponds to a total of 5200 milling heads on the wafer.

An overview of the mask design can be seen in Figure 2. The left side shows the conceptual design on a chip. The milling structures are arranged in a grid of one millimeter. Test structures consisting of an arrangement of beams are present on the left side with various widths. Alignment marks for grinding are placed on the outer contours in the shape of a cross. These marks define the corners of each individual chip. The alignment marks in between these crosses are used to grind a base area of one square millimeter with a central positioning in the milling geometry. On the right side, there is the contour of the planned milling head, which is available in diameters of 200 µm, 100 µm, 75 µm and 50 µm. The cutter structures have straight cutting edges and rounded outer sides. Alternatively, other geometries can also be created. However, the selected structure represents the most suitable solution with regard to the milling behavior [14].

Photolithography is used to create individual structures on the substrate according to the image of the designed mask. The photosensitive resist used is the AZ 10 XT [15] from MicroChemicals. This positive resist achieves a resist thickness of 10 μm with standard processing [15].

### 2.2. Electroplating

Nickel is deposited by means of electroplating (Figure 1iii), filling the exposed structures. Nickel masking is used due to its high resistance in fluorine gas atmospheres to achieve high selectivity during reactive ion etching [16].

The wafer is placed in the nickel electrolyte NB 100 from MicroChemicals. A DC power source is used for the deposition. The wafer is contacted as the cathode and a nickel electrode as the anode. For optimal deposition, a bath temperature of 30 °C is required [17]. The process current is regulated by the power source and is dependent on many parameters.

Equations (1) and (2) are used to calculate the process time and process current [18]. Equation (1) for calculating the process time *t* includes the molar mass M and the density ρ of the material to be deposited. In addition, the planned layer thickness *h*, the electrically conductive area *A*_*el*_*,* the valence of the dissolved ions of the metal to be deposited *z* and the Faraday constant *F* are also taken into account.
(1)$$\text{𝑡}=\frac{h\;\times \;A_{el}\;\times \;\rho \;\times \;z\;\times \;F}{I\;\times \;M}$$
*I* = *β* × *A*_*el*_(2)

To calculate the process current, the specific surface current density *β* of the electrolyte is multiplied by the electrically conductive area *A*_*el*_*,* as shown in Equation (2). The specific surface current density *β* of the used nickel electrolyte is 20 mA/cm^2^. For the calculation, additional information on the surface area and the planned coating thickness is required. The deposition is followed by a rinsing and cleaning step. Afterwards, the structures are checked for defects using an optical microscope, and the reactive layer thickness is then measured using a tactile profilometer. Finally, the resist is removed (Figure 1iv) in acetone.

### 2.3. Ion Beam Etching

The exposed areas of the overall metallization are removed by means of ion beam etching (Figure 1v). In this process, the beam voltage is 750 V, the beam current is 100 mA and the process gas flow is 12 sccm argon. The process time is determined via a visual inspection during the running process and ends when no metallic shine is visible on the wafer surface. A resistance test after the process using a multimeter indicates the remaining metallization on the surface. Ion beam etching is successfully carried out. The process time is 11:30 min, which is confirmed by a resistance measurement.

### 2.4. Reactive Ion Etching

In the process of reactive ion etching (Figure 1vi), the milling contour is etched. The existing nickel structures serve as an etching mask. The main focus of the investigations is the reactive ion etching process, in which an optimal process will be identified. A high etch rate in connection with structurally accurate milling cutter geometries are the target values. One measure of structural fidelity is, among others, anisotropy. The calculation of the degree of anisotropy r can be calculated with Equation (3) [19]. Here, *a_v_* stands for the vertical etch removal and *a_h_* for the horizontal removal. A value of r = 1 indicates purely anisotropic etching behavior. A value of r = 0 reflects purely isotropic etching behavior. In reality, a value between zero and one is most common, as it is then a mixed form.
(3)$$r=\frac{a_{v}-a_{h}}{a_{v}}$$

A parameter analysis is carried out to investigate the optimal range of parameters. The design of experiments (DOE) is performed using a 2-stage full factorial experimental design [20]. One central point per block is determined. The replications of the corner points amount to one. The five factors used are composed of the temperature, power, pressure, and the gas flows of oxygen and sulfur hexafluoride [21]. In the optimal target range, the resolution of the factors is increased in order to determine the target value more precisely. The dependencies among the factors are determined up to the 2nd degree. An overview of the parameters used can be seen in Table 1. For the gas composition, the ratio between the oxygen and sulfur hexafluoride is being studied. Another parameter is the temperature, which cannot be set directly but is realized via the implemented idle times in the process.

The series of tests begins with pure SF_6_ plasma in which the oxygen content is increased in two stages. The pressure during the process is controlled by a butterfly valve, which is operated manually. An operating pressure of 8 × 10^−4^ mbar is achieved when the butterfly valve is fully closed, and it assumes a pressure of 9 × 10^−5^ mbar when the valve is open. A value of 45 min is set as the process time in order to etch a removal in the range of several microns and to draw conclusions about the edge steepness and the grade of anisotropy. The removal and grade of anisotropy are visually measured using secondary electron microscopy (SEM) and confocal laser scanning microscopy (CLSM).

### 2.5. Wet Chemical Etching

After the dry etching process, the wet chemical etching (Figure 1vii) follows. Aqua regia is used to remove the remaining nickel masking with the associated seed and adhesive layers of permalloy and chromium. The silicon carbide is inert to aqua regia, which is a mixture of three parts concentrated hydrochloric acid and one part concentrated nitric acid. The process time is determined by means of visual inspection. The etching success is checked under a light microscope. After a process time of 16 min, no metallization is remaining. After processing, the chip is rinsed in DI water.

### 2.6. Back-End Processes

The penultimate step is the separation of the structures via dicing to a square base with an edge width of one millimeter (Figure 1viii). The cutter structure is positioned centrally on the base surface. For separating the chips of the silicon carbide wafer, the dicing machine Disco DAD 3350 is used. Therefore, a protective resist is applied before to protect the structures. This is the AZ 10 XT, which is processed with the standard parameters (see Table 1). The protective coating also offers the advantage of simultaneously removing the impurities produced during dicing during subsequent removal with acetone. The wafer is then fixed in place with blue tape. Using an optical alignment system, the release marks on the wafer can be located and aligned.

A resin-based hubless diamond abrasive blade is used, which has a grit size of 15 microns with a blade width of 100 μm. Since silicon carbide is a hard material with a Mohs hardness of 9.6, some adjustments have to be made in the dicing process compared to standard processes used for pure silicon. The infeed as well as the feed rate must be kept low. The feed rate can be estimated at 0.5 mm/s. The total infeed of the wafer thickness is achieved in two equal partial steps in order to keep the wear of the grinding blade low. The process takes place at a rotational speed of 20,000 rpm and cooling water supply. In addition, a light microscope examination provides information about chipping that occurs during dicing.

Finally, the manual joining process for the milling head and milling shaft is carried out using UV-initialized adhesive (Figure 1ix). Adhesive bonding combines many advantages, but nevertheless places many demands on the adhesive to be used.

These requirements are met by the UV 2133 adhesive from Polytec PT. This is a UV-curing adhesive on a methacrylate hybrid basis. Table 2 lists the most important parameters of the adhesive. The adhesive is characterized by very high chemical and moisture resistance, with a high thermal conductivity of 0.6 W/mK, which confirms the adhesive for the intended application [22].

Degreasing is required as a surface treatment for the two joining partners. Alternative processes such as eutectic bonding or soldering are not suitable due to the long process times and high temperatures. Due to its UV-initialization, the UV-adhesive enables good process control as well as short holding times.

The manual process is performed during joining. By irradiating the joint with light of a wavelength of 395 nm for 2 seconds, the curing process is completed and the final strength of the adhesive is achieved. The bond is evaluated in the shear tester. Shear strengths of at least 10 MPa should be withstood. The planarity and concentricity of the joint are checked in CLSM.

The complete process plan is available in Appendix A in Table A1.

### 2.7. Milling Tests

After completion of the milling tools, a function test with regard to the tool life, wear and width of the milling contour is carried out. Face milling is used for the milling test. The travel path is planned in such a way that the greatest possible travel can be achieved over a small area, which is realized via a meander structure. The planned travel of the milling cutter is 1500 mm due to the limited working space. In order to determine whether the occurring wear patterns have general validity or are of a material-specific nature, milling tests are carried out on the materials copper and aluminum, which are non-alloyed, and a C45-steel. In all the materials, a spindle speed of 45,000 rpm is used. This value represents the possible maximum realized in the used machine tool. With this limited spindle speed, the copper and aluminum are processed at cutting speeds of 15 m/min and the steel at 7.2 m/min, which are common cutting speeds in the materials. The milling processes take place with the use of cooling lubricant.

Most of the milling tests are conducted in copper. The values used for the feed and infeed correspond to common values of conventional milling cutters of the diameter. A z-feed of 5 µm is performed. Before the milling test, planarization is carried out using an 8 mm milling cutter and the depth axis is calibrated by means of surface scanning.

Aluminum is used to investigate the machinability in other materials, as it is a light metal that is often used in industry as a substitute for steel. Due to the easily machinable character of the material, the same process parameters are chosen for the milling process. Before the milling test, planarization is also carried out using the same cutter as copper.

For the milling test in steel, the surface of the workpiece is milled flat in preparation. This is done with a coated milling cutter with a diameter of 3 mm.

## 3. Results

In this section, the results from the process development are presented and discussed. First of all, the process for manufacturing the micro-milling cutters is classified, followed by the results of the application in milling tests.

### 3.1. Metallization and Photolithography

The sputtering step (Figure 1i) achieved the specified layer thicknesses. It is advisable to process and apply the electroplating promptly, as longer waiting times can lead to adhesion problems between the starting layer and the electrodeposited layer. The reason for this could be the formation of a native oxide layer on the permalloy layer. A possible solution for longer process interruptions is storage in an exicator.

After each process step of photolithography, a visual inspection with a light microscopy is done. The applied resist thickness is checked with a tactile profilometer. The targeted layer thickness of 10 µm is achieved on the entire wafer. A layer thickness of 10.7 µm is found in the center of the wafer, with an average of 10.3 µm in the edge areas. After development, an optical check under the light microscope must be carried out to record the progress of the development.

### 3.2. Electroplating

The used masked concept results in a surface area to fulfill of 19.87 cm². A theoretical layer thickness of 10 µm is aimed. After the calculated process time of 73 min, the deposition is measured with tactile profilometry. The nickel layer is 9.6 µm, which results in a yield of 96% compared to the calculated value.

### 3.3. Reactive Ion Etching

The tests for reactive ion etching are carried out with the specified parameters from Table A2. The tests are performed once, although a repetition would allow a statement about the reproducibility, which is planned in further work. Subsequently, each sample is examined via SEM with regard to ablation and anisotropy. After analyzing the measurement results, the correlations between the individual parameters can be established. No significant dependencies between the individual factors could be evaluated.

Due to the now visible dependence on the parameters, the etching behavior can be compared with the ideal values from the literature with regard to the gas composition. The gas ratio has a different dependency on the etch rate for silicon carbide. The ideal ratio between SF_6_ and O_2_ is given as 5:1 [16,18], and this range is to be represented by supplementary experiments. This is done in experiments 18–26 of the parameter analysis, in which the oxygen flow assumes values of 2 sccm and 5 sccm. These flow rates enable the gas ratios between 6:1 and 2.4:1. The greatest removal can be found at the O_2_-flow values of 2 and 3 sccm. The excerpt of the generated values is listed in Table 3.

To check for a further increase, a test is carried out at 2.5 sccm O_2_ and known parameters. Due to the limited control of the flow controller used, only an adjustment in steps of 0.5 sccm is possible. The basis for this is an interpolation of the behavior based on already recorded values, which is seen in Figure 3.

The postulated increase to an ablation value of 4.54 μm and, correspondingly, an achieved etch rate of 0.1 μm/min is shown. A seal change in the RIE system has reduced the contamination of the process gases with nitrogen, resulting in an increase in the etch rate to 0.34 µm/min. With the optimal values, the gas ratio of the process gases of sulfur hexafluoride and oxygen is 5:1, which confirms the results from the literature.

There is a trend for the etch rate to increase with increasing RF power or process pressure. When the power is doubled, the etching rate increases by a factor of 1.04 ± 0.1. An increase in the process pressure by a factor of 5.5 leads to an increase in the etch rate by a factor of 13 ± 5. A scheme for these trends is shown in Figure 4.

#### Influence of Temperature on Anisotropy

The isotropy behavior is still prevalent in the RIE process and has so far been sufficient for only the limited use of the cutter structures for later use as milling cutters. From tests already carried out in machine optimization, the temperature has proved to be the key factor in achieving high anisotropy. The bias voltage that occurs results in increased physical erosion and an increase in anisotropy with increasing values. This is limited due to the system conditions of the RF generator. The maximum bias voltage value is 850 V.

Lowering the substrate temperature causes an increase in anisotropy. This relation is visualized in Figure 5.

Figure 6 shows an SEM image of the test structures. A high underestimation of the structures can be seen, which amounts to a degree of anisotropy of 0.8. Structures with a diameter smaller than 200 µm cannot be used due to their changed shape. The top area is separated due to the tapering in the middle. The following process steps are performed with milling heads of 200 µm diameter.

The developed process parameters of the RIE process are presented in Appendix A Table A2. A power of 600 W with the butterfly valve closed is specified. An O_2_ flow of 2.5 sccm and an SF_6_ flow of 12 sccm give the process gas ratio. These parameters result in a large bias voltage of 850 volts, achieving the highest anisotropy with an anisotropy factor of 0.8 and an etching rate of 0.34 µm/min of the silicon carbide. The RIE process shows the highest etch rate and anisotropy at the values listed in Table 4.

The micro-milling heads have an average structure height of 66 ± 2.4 μm and a grade of anisotropy of 0.8. This results in a decreased diameter of 160 µm at the tip of the milling structures. The achieved structure heights offer the possibility of milling with common infeed depths of a few micrometers for micro-machining. Conventionally available milling cutters have an aspect ratio of 1:2, which represents an optimization envelope. The anisotropy of 0.8 is reflected in the milling track. To achieve vertical flanks, the anisotropy must be increased. When using smaller structure sizes, the effect of undercutting is too high for use as a milling cutter.

### 3.4. Back-End Processes

The diced milling heads can be seen in Figure 7. Chipping occurs in the range of a maximum of 12 µm along the dicing street, which does not affect the functionality of the chips. The cutter structures are still intact after the dicing and stripping of the protective resist.

A manual joining is performed. The curing time of two seconds is held for each cutter. In the tests, shear strengths greater than 10 MPa are consistently maintained. With a test number of ten cutters, an average shear strength of 13.7 ± 1.2 MPa is achieved. The tilt and offset to the center point are examined in CLSM. Both are in the range of 20–30 µm. With cutter diameters of 200 µm, this tolerance is acceptable, but with further miniaturization, these tolerances become more influential. This is reflected in a faulty milling track and greater wear. The manually performed joining process can be converted into an automated process with a pick-and-place machine, which may result in reproducible and optimal properties with regard to planarity and concentricity.

### 3.5. Milling Tests

After the process development, milling tools with a structure diameter of 160 μm are successfully produced. Figure 8 shows an SEM image of a completed milling cutter with a structure diameter of 160 μm. The joint between the milling head and milling shaft is clearly visible, which has a rather rough appearance with regard to the dimensions of the cutter head due to the manual process.

In all the milling tests, there is wear of the milling cutter but not failure, except one trial in the aluminum, which means that the defined tool life of 1500 mm is maintained by ten milling cutters. The tool life of the micro-mills represents a minimum given by the travel path in the material. By planning a different travel path, the maximum tool life can be evaluated. A rounding of the cutting edges after the milling test can be seen. This is localized where the highest forces occur in the milling process.

In Figure 9, SEM images of the milled tracks in the copper, aluminum and steel are shown in (a). Additionally, a detailed image of the associated milling head after the milling operation is provided in (b). In Figure 9b, possible wear behavior is pictured.

In copper, a frequently remaining burr in the milling track is recognizable in comparison to the other two materials. The milled track is characterized by a homogeneous image along the milled track. At the transition into the depth, there are no perpendicular flanks, although there is the presence of an angle.

Looking at the milling head after the milling process in aluminum, there is a failure pattern. The structure shows almost complete shearing of the cutter head from the substrate base. A possible reason for this effect is the good solubility of the silicon components of the ceramic in the aluminum material, which can lead to chemical reactions in the component and to premature failure [10,13]. The occurring mechanical stress could be another reason for failure [13]. These represent possible hypotheses to explain the wear on the cutter head. In future tests, confirmation of this fact must be evaluated.

The same wear behavior can be observed in steel as in copper. In steel, finer grooves are noticeable due to the lower feed rate.

The characteristic of angled flanks can be found in all the milled tracks. The reason is the isotropic etching behavior in the RIE process, which causes a reduction of the structure from the substrate base to the cutter tip (cf. Figure 3). A comparison of the milling tracks of different cutters reveals a difference in the width of the milling track, which ranges from 97 μm to 252 μm for a 200 µm diameter milling tool.

When looking at the various cutter heads, there is no difference in the cutter diameter. Both the tilt and coaxiality are responsible for the different widths, which cause the cutter to rotate outside the optimal axis.

The milling heads always show a similar wear pattern. Often there is no wear up to a rounding of the cutting edges. Due to the small number of milling tests, no reliable statement can be made about the material-related wear in copper. Due to a delocalization of the milling heads, wear is conceivable because stronger forces act on the cutting edge due to the asymmetrical and one-sided engagement and thus lead to premature wear.

With the exception described above, the milling tools remain in an intact condition after the milling process. Considering all the wear phenomena of the milling heads, no material-specific wear can be determined. This must be evaluated by future tests in the materials in order to be able to make a statistical statement.

After comprehensive evaluation of the process and the milling tests, optimization potential can be identified in many areas, which can be explored in future tests. The number of tests only allows a qualitative statement about the use of the milling cutters. Further tests for the quantitative classification and the verification of the reproducibility are planned. The tests carried out serve to make an initial statement on the suitability for use.

The use of the dry etching process for the cutter structures is mainly characterized by the attribute of high anisotropy greater than 0.8. The improvement of the cutters with regard to a higher edge steepness in order to obtain a better milling track geometry is at the core of the outlook.

## 4. Conclusions

This paper shows the process development for the batch production of micro-milling tools that are structured photolithographically and manufactured via reactive ion etching.

A photolithographic process was chosen as the basis for the overall process. First, a glass mask with various cutter diameters from 50 to 200 µm was designed. The process sequence enables batch production of 5200 milling cutters in variable geometries. In this photolithographic process, a metal masking was selected as the etching mask for the reactive ion etching. An RIE process was developed showing the highest etch rate of 0.34 µm/min and an anisotropy factor of 0.8 at the values listed in Table 4.

The masking was generated by means of electrodeposition. Dry etching was followed to generate the milling contours. Dissolution via wet chemical of the metal masking was followed by the separation and joining of the milling structures.

The functionality of the milling cutters was evaluated in the final milling tests. The milling cutters had a tool life of at least 1500 mm. The cutting speeds were 15 m/min for copper and aluminum and 7.2 m/min for steel at a spindle speed of 45,000 rpm. The milling tracks had different track widths, which tapered with increasing depth. The tapering was due to the partly isotropic etching profile of the structures.

The varying widths of the milled track were the result of the joining process, in which the milling shank and milling head were brought together, as well as the decentralized positioning during the cut-off grinding. In this manually performed process, there was a tilting and coaxiality of 20 to 30 µm of the cutter heads. As a result, there were also higher cutting forces on the cutting edges on one side, which explained the wear patterns that occurred. In all the tests, the UV-curing acrylic adhesive joint kept functionality.

The named goals for the development of a novel process for the manufacture of micro-milling tools by means of dry etching and the verification of their function could be achieved and transferred into an alternative workflow to conventional milling cutters. ICP technology enables higher structural fidelity due to improved structural flanks with reduced undercutting. This in turn ensures further miniaturization of the structure diameters and a defined structure transfer in the milling track. The results demonstrate that a large influence on the milling success is demanded from the back-end processes. One possibility for optimization is the automation of the joining process on suitable equipment such as a pick-and-place machine with an optical adjustment.

All the aforementioned aspects show the usability of the developed process and show great potential for optimization and room for innovation. Thus, in the future, it may represent a more economical production method for the manufacture of micro-milling tools by means of a dry etching process in a batch production and a competitive alternative to conventional manufacturing. The advantages of the manufactured micro-milling tools consist of the reduction of manufacturing costs due to batch production. This results in an increase in economic efficiency. A further scaling of the chip dimensions is reflected in a further cost reduction per chip. Another advantage is the freedom of design of the cutting geometry, which is defined in the photolithographic process. Further layout options can be implemented in the future by means of gray tone lithography. Furthermore, the cutter is made entirely of silicon carbide, which promises a longer milling operation than a cutter with a coating. Possible disadvantages can be found in the joining process, on whose tolerances the milling success depends. In addition, conventional micro-milling cutters have spiral cutting edges, which improve the milling result and chip removal. This geometry cannot be implemented in the developed process. Further statements on the limits of the application behavior and the process cannot be determined with certainty due to the limited number of tests.

In the application, the optimum surface qualities must be achieved and the machining of demanding geometries as well as the dimensional accuracy of the workpiece must be guaranteed. The photolithographic process sequence allows tolerances in the range of a few micrometers and ensures these requirements. Possible applications include narrow cavities, rib geometries, bores and small corners for use in micro-fluidics and the manufacture of lenses. In these areas, production by means of complex photolithography in a clean room environment can be replaced by machining.

This work provides a promising outlook for further research with the prospects already mentioned and establishes the foundation for a new production strategy in the field of milling cutter manufacture.

## Figures and Tables

**Figure 1 micromachines-14-00580-f001:**
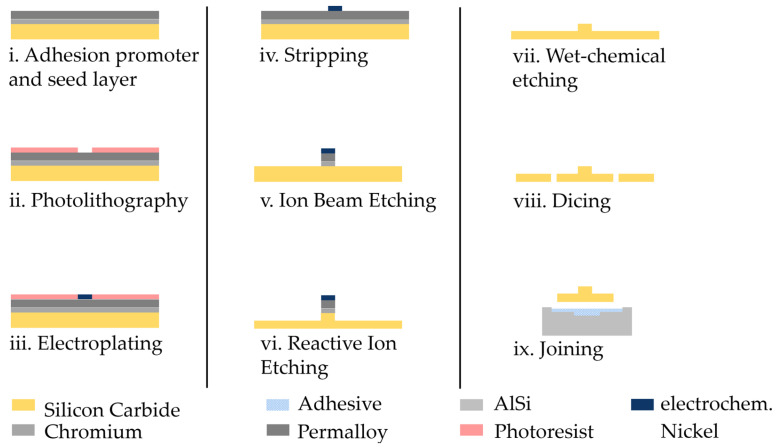
Diagram of the manufacturing process (not to scale).

**Figure 2 micromachines-14-00580-f002:**
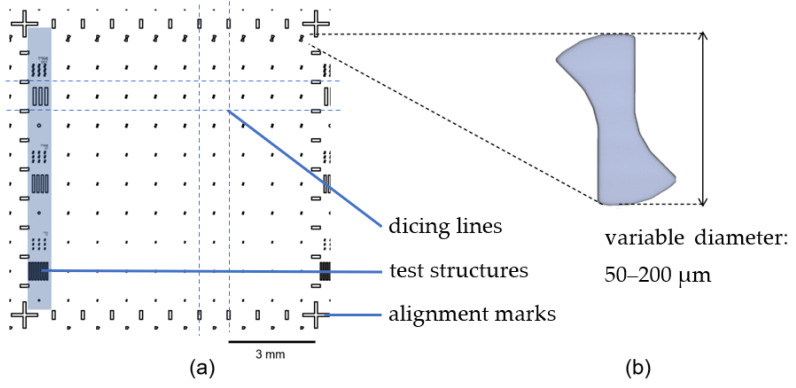
Concept of the mask: (**a**) an excerpt of 10 × 10 mm^2^ of the mask concept and (**b**) detailed image of the milling geometry with various diameters.

**Figure 3 micromachines-14-00580-f003:**
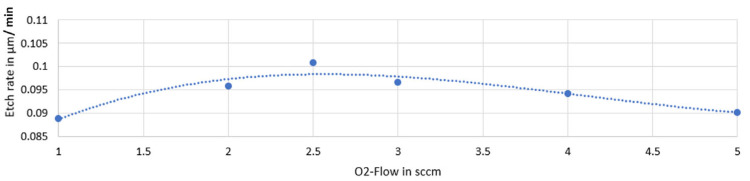
Relation between the O_2_ flow and etch rate.

**Figure 4 micromachines-14-00580-f004:**
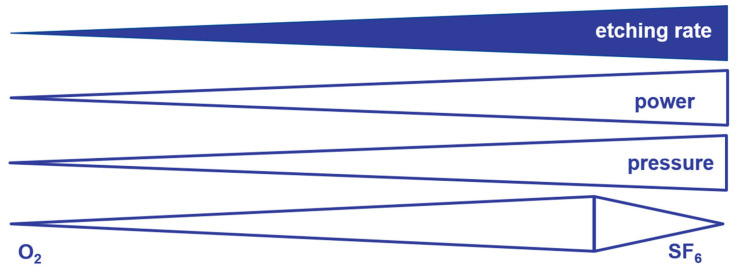
Influence of process parameters on the etch rate.

**Figure 5 micromachines-14-00580-f005:**
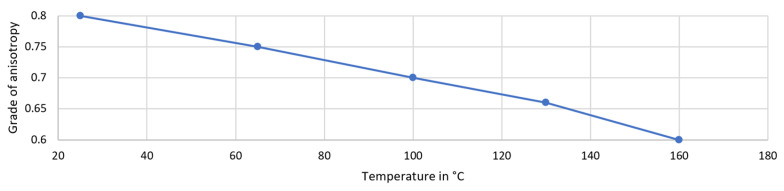
Influence of temperature on anisotropy.

**Figure 6 micromachines-14-00580-f006:**
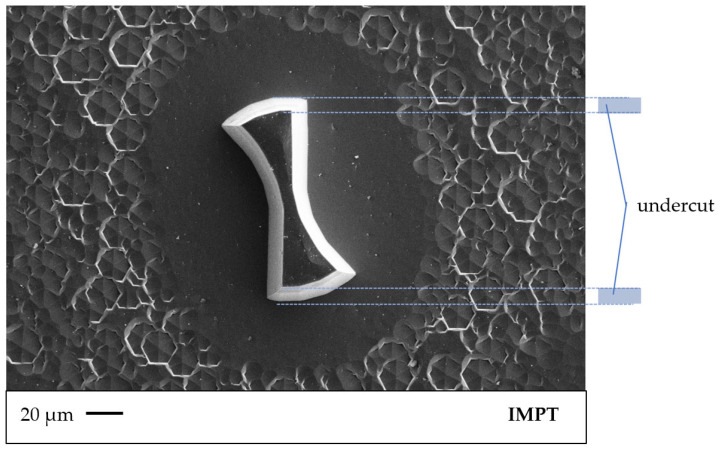
Underestimation of the structures; marked in blue is the effect of isotropy.

**Figure 7 micromachines-14-00580-f007:**
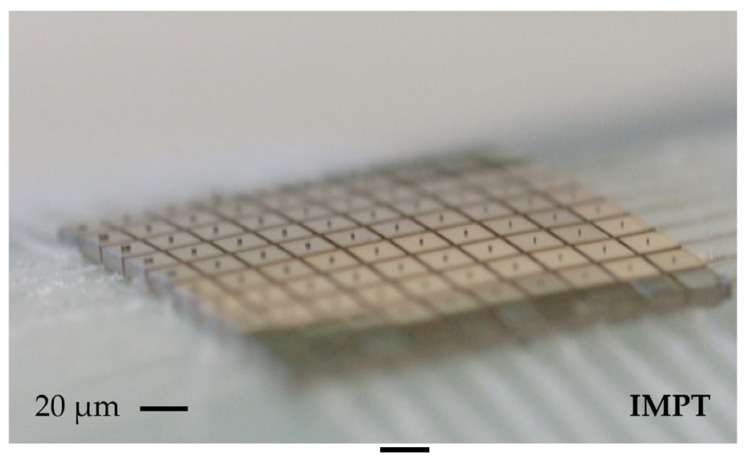
Diced chips on adhesive tape.

**Figure 8 micromachines-14-00580-f008:**
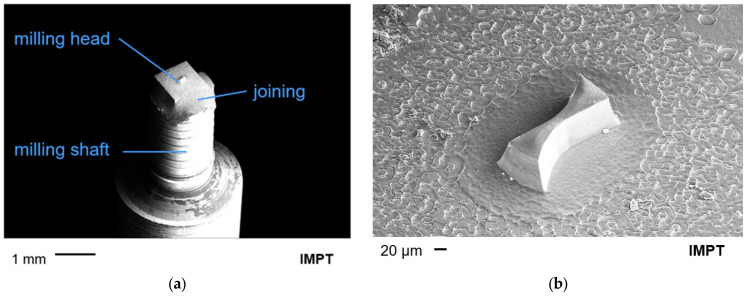
SEM image of (**a**) finalized milling tool with a diameter of 160 µm and (**b**) a detailed view.

**Figure 9 micromachines-14-00580-f009:**
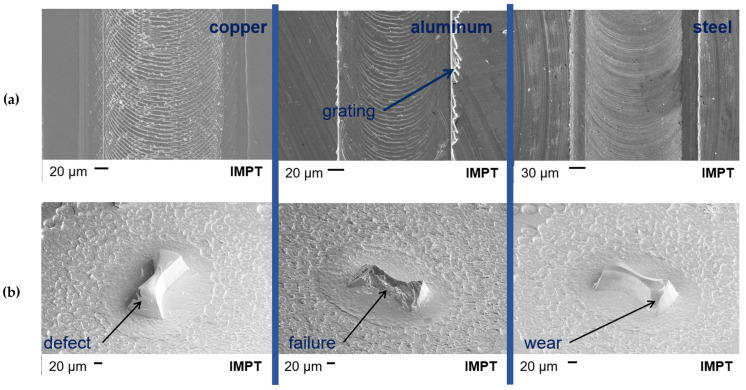
Shown is (**a**) an SEM image of a milled track in copper and (**b**) a detailed image of the milling head after the milling operation.

**Table 1 micromachines-14-00580-t001:** Parametric analysis: range of studied parameters.

Parameter	Range	
	Minimum	Maximum
SF_6_ flow	4 sccm	12 sccm
O_2_ flow	0 sccm	10 sccm
Power	300 W	600 W
Pressure	0%	100%
Temperature	No cooling	1:1 Process time: idle time

**Table 2 micromachines-14-00580-t002:** Parameters of Polytec PT UV 2133 [22].

Base	Density [g/cm^3^]	Viscosity [mPa·s]	Max. Temperature [°C]	Hardness [Shore]
Methacrylate-hybrid	1.60	35,000	160	80D

**Table 3 micromachines-14-00580-t003:** Excerpt of the results of the RIE parameter analysis.

Trial	O_2_-Flow [sccm]	Gas Ratio [SF_6_:O_2_]	Removal [µm]
21	1	12:1	4.0
22	2	6:1	4.3
23	3	4:1	4.4
24	4	3:1	4.2
25	5	2.4:1	4.0

**Table 4 micromachines-14-00580-t004:** Parametric analysis: optimized parameters.

Parameter	Optimized Values	
SF_6_ Flow	12 sccm	
O_2_ Flow	2.4 sccm	
Power	600 W	
Pressure	5 × 10^−4^ mbar	
Substrate Temperature	20 °C	

## Data Availability

The datasets generated and analyzed during the current study are available from the corresponding author on reasonable request.

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
