# Peer review of "Process Development for Batch Production of Micro-Milling Tools Made of Silicon Carbide by Means of the Dry Etching Process"

_micromachines, 2023, doi:10.3390/mi14030580_

Round 1

Reviewer 1 Report

This paper studied a process to manufacture the silicon carbide micro milling heads in a batch production. The milling tools are manufactured by reactive ion etching with structure diameters down to 50 μm. The following comments are given for considerations.

1)Fig.3, Fig.5, the data was not averaged for multiple times (at least three times) and the error bar was not indicated. It is recommended that the author supplement the experiment and insert the error bar.

2)  In Sction 3.5, the reason for the wear of silicon carbide when machining aluminum was proved by listing the corresponding literature. The literatures should be given and discussed to support the arguments.

3) The process to manufacture micro-milling tools by means of dry etching and the verification of their function is argued to be able to transfer to an alternative workflow to conventional milling cutters. However, only a simple SEM diagram was presented in the final milling tests. The tool life and other milling tests should be added to support the applications.

4)The conclusion part of this article is too long. Please reduce it.

Author Response

Good day dear reviewer,

Thank you for the constructive comments and their explanation. These have helped us a lot in revising the work.

1)Fig.3, Fig.5, the data was not averaged for multiple times (at least three times) and the error bar was not indicated. It is recommended that the author supplement the experiment and insert the error bar.

1)  Comment revised: Revised information on how to conduct the experiment and made adjustments regarding the informative value for reproducing the measurements l.: 178-187; 292-297

2)  In Section 3.5, the reason for the wear of silicon carbide when machining aluminum was proved by listing the corresponding literature. The literatures should be given and discussed to support the arguments.

2)  Comment revised: Added literature and section revised l.:396-402

3) The process to manufacture micro-milling tools by means of dry etching and the verification of their function is argued to be able to transfer to an alternative workflow to conventional milling cutters. However, only a simple SEM diagram was presented in the final milling tests. The tool life and other milling tests should be added to support the applications.

4)  Comment revised: Milling results revised and quantitative statements added l.: 383 f.

4)The conclusion part of this article is too long. Please reduce it.

4)  Comment revised: Reduction and rearrangement of the conclusion, including of advantages/disadvantages and use l. 433 f.

Best Regards,

Christian Wittek

Reviewer 2 Report

In general, the manuscript is well written. Suggestions for improving the manuscript are as follows:

1. The Abstract should contain answers to the following questions: What problem was studied and why is it important? What methods were used? What are the important results? What conclusions can be drawn from the results? What is the novelty of the work and where does it go beyond previous efforts in the literature? Please include specific and quantitative results in your Abstract.

2. At the end of the Introduction section, emphasize the scientific contribution of your research. Also write scientific hypotheses.

3. The authors emphasize that they performed a parameter analysis in order to obtain the optimal range of parameters. It is not clear which optimization method they used and how they obtained the optimal parameters.

4. Authors should explicitly write the advantages and disadvantages of your micro milling tools compared to others (possibility of micro machining, costs, etc.)

5. Potential errors should be further analysed and discussed. Can measurement uncertainty be estimated?

6. In addition, analyse and discuss the possibilities of practical application.

7. I think the Conclusion section should be corrected, supplemented and updated. The results should be displayed in the Results section. In the Conclusion section, only the main results should be presented, and it is necessary to additionally emphasize: scientific contribution, innovative methodology, future research and the like. Also in the Conclusion section, the limitations of the application of the methodology should be emphasized (every research has advantages but also limitations).

Author Response

Good day dear reviewer,

Thank you for the constructive comments and their explanation. These have helped us a lot in revising the work.

1. The Abstract should contain answers to the following questions: What problem was studied and why is it important? What methods were used? What are the important results? What conclusions can be drawn from the results? What is the novelty of the work and where does it go beyond previous efforts in the literature? Please include specific and quantitative results in your Abstract.

1. Comment revised: included and revised abstract l.: 9-25

2. At the end of the Introduction section, emphasize the scientific contribution of your research. Also write scientific hypotheses.

2. Comment revised: included and revised introduction l.: 70-73

3. The authors emphasize that they performed a parameter analysis in order to obtain the optimal range of parameters. It is not clear which optimization method they used and how they obtained the optimal parameters.

3. Comment revised: described methods and procedure of parameter analysis, boundary conditions set l.: 178-187

4. Authors should explicitly write the advantages and disadvantages of your micro milling tools compared to others (possibility of micro machining, costs, etc.)

4.  Comment revised: section regarding advantages and disadvantages added; classification of the milling cutters to conventional manufactured milling cutters l.: 464-485

5. Potential errors should be further analysed and discussed. Can measurement uncertainty be estimated?

5. Comment revised: limitations of the process recorded and discussed

6. In addition, analyse and discuss the possibilities of practical application.

6. Comment revised: mentioned part added l.: 497-485

7. I think the Conclusion section should be corrected, supplemented and updated. The results should be displayed in the Results section. In the Conclusion section, only the main results should be presented, and it is necessary to additionally emphasize: scientific contribution, innovative methodology, future research and the like. Also in the Conclusion section, the limitations of the application of the methodology should be emphasized (every research has advantages but also limitations).

7.  Comment revised: updated conclusion, conversion of individual text sections, extension with previously mentioned topics l.: 433-488

Best Regards,

Christian Wittek

Round 2

Reviewer 2 Report

The manuscript has not been fully revised, but key points have been updated and supplemented, so I recommend accepting the manuscript in its current form.

Author Response

Thank you very much for your reviewing comments.